# *Quercus glauca* Acorn Seed Coat Extract Promotes Wound Re-Epithelialization by Facilitating Fibroblast Migration and Inhibiting Dermal Inflammation

**DOI:** 10.3390/biology13100775

**Published:** 2024-09-28

**Authors:** Shin-Hye Kim, Hye-Lim Shin, Tae Hyun Son, So-An Lim, Dongsoo Kim, Jun-Hyuck Yoon, Hyunmo Choi, Hwan-Gyu Kim, Sik-Won Choi

**Affiliations:** 1Forest Biomaterials Research Center, National Institute of Forest Science (NIFoS), Jinju 52817, Republic of Korea; black7a@korea.kr (S.-H.K.); hlims0901@korea.kr (H.-L.S.); snoopyegg@korea.kr (T.H.S.); skimds@korea.kr (D.K.); jhyoon7988@korea.kr (J.-H.Y.); 2Department of Biological Sciences, Jeonbuk National University, Jeonju 54896, Republic of Korea; hgkim@jbnu.ac.kr; 3Pharmacogenomics Research Center, Inje University College of Medicine, Busan 47392, Republic of Korea; limsa0223@naver.com; 4Department of Forest Bioresources, National Institute of Forest Science (NIFoS), Suwon 16631, Republic of Korea; choihyunmo@korea.kr

**Keywords:** *Quercus glauca* Thunb., acorn seed coat, wound healing, skin damage, antioxidant

## Abstract

**Simple Summary:**

This research investigates the potential therapeutic applications of *Quercus glauca* acorn seed coat water extract (QGASE) in the context of wound healing. The results indicate that QGASE significantly enhances wound closure in human dermal fibroblasts by upregulating critical markers associated with the wound-healing process. Furthermore, QGASE demonstrates antioxidant properties that mitigate oxidative stress and promote recovery from injuries induced by hydrogen peroxide, thereby providing evidence of its anti-inflammatory effects. In vivo experiments further corroborate the efficacy of QGASE in facilitating wound healing. The results of our study enhance the existing body of literature regarding phytotherapy and its potential applications in the field of wound care. This research seeks to identify a promising candidate for the development of innovative therapeutic strategies, thereby addressing the urgent need for safe and effective wound-healing therapies.

**Abstract:**

The skin, recognized as the largest organ in the human body, serves a vital function in safeguarding against external threats. Severe damage to the skin can pose significant risks to human health. There is an urgent requirement for safe and effective therapies for wound healing. While phytotherapy has been widely utilized for various health conditions, the potential of *Quercus glauca* in promoting wound healing has not been thoroughly explored. *Q. glauca* is a cultivated crop known for its abundance of bioactive compounds. This study examined the wound-healing properties of *Quercus glauca* acorn seed coat water extract (QGASE). The findings from the study suggest that QGASE promotes wound closure in HF cells by upregulating essential markers related to the wound-healing process. Additionally, QGASE demonstrates antioxidant effects, mitigating oxidative stress and aiding in recovery from injuries induced by H_2_O_2_. In vivo experiments provide additional substantiation supporting the efficacy of QGASE in enhancing wound healing. The collective results indicate that QGASE may be a promising candidate for the development of innovative therapeutic strategies aimed at enhancing skin wound repair.

## 1. Introduction

The skin is the largest organ in the human body and serves a vital function in safeguarding internal tissues from a range of external factors, including chemicals, pathogens, and UV radiation. Severe skin injuries have the potential to be life-threatening [1]. The wound-healing process is characterized by four distinct stages: hemostasis, inflammation, proliferation, and remodeling. While these stages follow a temporal sequence, they also exhibit overlapping characteristics [2]. Wound healing is a multifaceted process characterized by complex interactions and responses among various cells and mediators [3]. Hemostasis is initiated through the activation of platelets, which subsequently leads to the release of various growth factors, including PDGF, TGF, FGF, and VEGF [4]. These growth factors play a crucial role in controlling cellular responses such as migration, differentiation, and proliferation during wound healing [5]. The inflammatory stage is initiated by acute signals, which include DAMPs and PAMPs. These signals are recognized by TLRs, which serve to initiate and sustain the inflammatory response. Neutrophils and macrophages release various pro-inflammatory cytokines and chemokines that attract fibroblasts and epithelial cells [6]. Angiogenesis is essential for epithelial cell proliferation, migration, and the formation of new vascular vessels [7]. Concurrently, re-epithelialization encompasses the proliferation of terminally differentiated epidermal cells, in addition to unipotent epidermal stem cells that originate from the basement membrane [8]. While the closure of acute and chronic wounds is generally regarded as the endpoint of wound healing in clinical practice, it is important to note that wounds may continue to experience tissue maturation or remodeling for several months or even years thereafter [7]. Consequently, the regulation of molecules implicated in the wound-healing process may present a promising strategy for safeguarding the skin against injury.

ROS have been recognized as crucial regulators of numerous biological processes, including inflammation, cell proliferation, angiogenesis, granulation, and the formation of the ECM throughout the progression of the wound-healing cascade [9]. Research indicates that even low levels of oxidative stress can function as chemoattractants for inflammatory cells [10]. Furthermore, research has demonstrated that H_2_O_2_ can activate the inflammatory response in both normal human cells and cancer cells [11]. It is essential to acknowledge that increased concentrations of H_2_O_2_ may result in delayed wound healing [12]. Consequently, an excess of ROS at the wound site may prolong the inflammatory phase, thereby hindering the wound-healing process [13]. Various molecular markers specific to distinct cell types play vital roles in wound healing. For instance, fibronectin is essential for interacting with other cells to facilitate ECM formation across all stages of wound healing [14]. Therefore, targeting the reduction of inflammation triggered by oxidative stress could serve as a crucial approach to enhance effective wound remodeling.

Phytotherapy, the practice of using plants for medicinal purposes, has a long-established history in treating various illnesses, including wound healing [15,16]. *Quercus glauca* Thunb., a medium-sized evergreen tree native to East Asia and belonging to the Fagaceae family, has been traditionally used in Chinese medicine for conditions like dysentery [17]. Phytochemicals, the bioactive compounds present in plants, exhibit promising therapeutic properties against a wide range of diseases [18]. Evergreen oak species (Quercus) have historically been employed in traditional medications for a diverse range of purposes, including the treatment of urolithiasis, tremors, hemorrhages, fevers, infections, the reduction of inflammation, and the alleviation of oxidative stress [19]. Research has shown that extracts derived from the stems and leaves of *Quercus glauca* Thunb. exhibit both anti-inflammatory and antioxidant properties [20,21]. While certain *Quercus* species have been employed in treating ailments like hemorrhoids, inflammation, jaundice, and tumors, their potential in wound healing has not been thoroughly explored [21]. This study investigates the wound-healing effects of *Quercus glauca* acorn seed coat water extract (QGASE) both in vitro and in vivo. The study aims to identify potential therapeutic agents for wound healing and inflammation by examining a plant component that is typically discarded.

## 2. Materials and Methods

### 2.1. Preparation of QGASE

*Q. glauca* acorn seeds were collected on 4 November 2020, from Jinju, Gyeongsangnam-do, South Korea. The identification of *Quercus glauca* acorns was formally conducted by Hyun-Jun Kim, a researcher at the Forest Medicinal Resources Research Center in Korea (voucher specimen: FMRC-201104A1-3). The acorns were washed with clean sterile water and air-dried at 50 °C for 3 days to remove moisture. Then, the seed coats of dried *Quercus glauca* seeds were separated from the others. The dried seed coats of *Quercus glauca* acorns were extracted utilizing double-distilled water in a shaker incubator at a temperature of 60 °C for a duration of 24 h. The crude extracts were subsequently subjected to centrifugation, filtration, vacuum evaporation, and freeze drying to yield a dry powder. The *Quercus glauca* acorn seed extract (QGASE) was then reconstituted using a solution of 30 mg/mL DMSO.

### 2.2. Reagents and Antibodies

Fetal bovine serum (FBS) originating from the United States and antibiotics (penicillin and streptomycin) were obtained from Gibco (Thermo Fisher Scientific; Waltham, MA, USA). TRIzol reagent was sourced from Invitrogen (Carlsbad, CA, USA). Dimethyl sulfoxide (DMSO) was procured from Sigma-Aldrich (St. Louis, MO, USA), while H_2_O_2_ was acquired from Duksan Science (Seoul, Korea). Antibodies targeting actin and horseradish peroxidase (HRP)-conjugated anti-mouse, anti-rabbit, and anti-goat antibodies were procured from Santa Cruz Biotechnology (Santa Cruz, CA, USA). Additionally, antibodies against fibronectin, COL1A1, α-smooth muscle actin (α-SMA), β-catenin, and vimentin were obtained from Cell Signaling Technology (Danvers, MA, USA). The antibody specific to endothelin-1 was acquired from Invitrogen, while the antibody against keratin 14 was sourced from BioLegend (San Diego, CA, USA).

### 2.3. Cells

HF cells were procured from Cefo Bio (Seoul, Korea). These cells were cultured in Dulbecco’s Modified Eagle Medium (DMEM), supplemented with 10% FBS and antibiotics. The cells were incubated at 37 °C in a 5% CO_2_ atmosphere using an incubator (Panasonic, Osaka, Japan; model MCO-170AIC-PK), with the culture medium being replaced every 2 to 3 days.

### 2.4. Migration Assay

A Radius™ 96-Well Cell Migration Assay Kit (Cell Biolabs, Inc., San Diego, CA, USA) was utilized to assess the effects of wound healing. According to the manufacturer’s protocols, the 96-well cell migration plate was pretreated with 100 μL of Radius™ Gel Pretreatment Solution per well. Following the washing procedure with the wash solution, HF cells were seeded at a density of 1 × 10^4^ cells per well in a 96-well cell migration plate and incubated for 24 h. Subsequently, after washing with a fresh medium, 100 μL of 1× Radius Gel Removal Solution was added to each well and incubated for 30 min. After another wash with a fresh medium, the cells were cultured for 24 h in serum-free media containing doses of 0.1, 0.3, 1, 3, and 10 µg/mL of QGASE. After the 24-h incubation period, the media were aspirated. The treated cells were then fixed in 10% formalin for 5 min, rinsed with distilled water, and stained with a cell stain solution. The area of the wound-healing was quantified using the ImageJ software (ver. 1.53k).

### 2.5. Cell Cytotoxicity Assay

A CCK-8 assay was conducted to evaluate the cytotoxic effects of QGASE on HF cells. The cells were seeded at a density of 1 × 10^4^ cells per well in a 96-well plate and cultured for one day with concentrations of 0.1, 0.3, 1, 3, and 10 µg/mL of QGASE. Following the incubation period, cell cytotoxicity was assessed utilizing the CCK-8 assay (Dojindo Molecular Technologies, Rockville, MD, USA). Subsequently, the cells were transferred into media containing the CCK-8 solution and incubated for a duration of 30 min. The optical density was subsequently measured at 450 nm using a spectrophotometer (SpectraMax iD3, Molecular Devices, Sunnyvale, CA, USA).

### 2.6. RNA Isolation and Quantitative Reverse Transcription Polymerase Chain Reaction

The HF cells were cultured at a density of 1 × 10^5^ cells/mL in a 6-well plate and incubated for a period of 24 h. To assess the effects of QGASE, the cells were pretreated with QGASE for 1 h at the indicated doses. We utilized 500 μM of H_2_O_2_ in a fresh medium to assess the expression of inflammatory genes. The total RNA was extracted from the HF cells after the experiment using TRIzol reagent according to the manufacturer’s protocol. The total RNA was also extracted from the skin tissue surrounding the wound using TRIzol, in accordance with the manufacturer’s protocol, following a 9-day experimental period. The tissue, immersed in TRIzol, was homogenized using sterile scissors and subsequently mixed vigorously with chloroform. After the addition of chloroform, the cell lysates were incubated on ice for 10 min. The lysates were then centrifuged at 15,000 rpm at 4 °C for 10 min, and the supernatants were transferred to a new tube. Following the addition of 2-propanol, the crude RNA was precipitated by centrifugation at 15,000 rpm at 4 °C for 10 min. The RNA pellet was washed with 75% ethanol in DEPC-treated water. The purity and concentration of the extracted RNA were assessed using a NanoDrop™ 2000 spectrophotometer (Thermo Scientific; Waltham, MA, USA). cDNA was synthesized from 1 μg of total RNA utilizing the RevertAid First Strand cDNA Synthesis Kit (Thermo Fisher Scientific; Waltham, MA, USA), in accordance with the manufacturer’s protocol for qRT-PCR. Primer pairs were generated using online Primer3 software (ver. 0.4.0) [22] (Table 1). An SYBR Green-based qRT-PCR was conducted utilizing the QuantStudio™ 5 real-time PCR System (Thermo Fisher Scientific) in conjunction with the PowerUp™ SYBR™ Green Master Mix (Thermo Fisher Scientific). All sample mixtures were performed in triplicate, and the data were analyzed using the 2–ΔΔCT method described by Livak and Schmittgen [23]. HPRT1 and β-actin were utilized as internal controls.

### 2.7. Western Blotting

Following the QGASE or vehicle treatment, the cells were lysed using radioimmunoprecipitation assay lysis buffer (Cell Signaling Technology) supplemented with protease inhibitors. After a 10 min incubation on ice, proteins were extracted from the supernatant through centrifugation at 15,000× *g* for 15 min. The protein concentration in the lysates was quantified using a detergent-compatible protein assay kit (Bio-Rad, Hercules, CA, USA). Subsequently, the proteins were subjected to sodium dodecyl sulfate polyacrylamide gel electrophoresis and transferred to polyvinylidene difluoride membranes (Merck Millipore, Darmstadt, Germany). The membranes were incubated with the following antibodies: mouse monoclonal anti-β-actin (sc-47778, Santa Cruz, CA, USA) at a dilution of 1:500; rabbit monoclonal anti-fibronectin (#26836, Cell Signaling Technology, Danvers, MA, USA) at 1:1000; rabbit monoclonal anti-COL1A1 (#72026, Cell Signaling Technology) at 1:1000; rabbit monoclonal anti-α-SMA (#19245, Cell Signaling Technology) at 1:1000; rabbit polyclonal anti-Keratin14 (905304, BioLegend, San Diego, CA, USA) at 1:1000; rabbit monoclonal anti-β-catenin (#8480, Cell Signaling Technology) at 1:1000; rabbit polyclonal anti-Endothelin-1 (PA3-067, Invitrogen) at 1:1000; and rabbit monoclonal anti-Vimentin (#5741, Cell Signaling Technology) at 1:1000. As secondary antibodies, mouse monoclonal anti-rabbit conjugated HRP (sc-2357, Santa Cruz) at 1:3000, mouse monoclonal anti-goat conjugated HRP (sc-2354, Santa Cruz) at 1:3000, and goat monoclonal anti-mouse conjugated HRP (sc-2031, Santa Cruz) at 1:3000 were utilized. Clarity Western ECL Substrate (Bio-Rad, Hercules, CA, USA) was employed for blot development, and visualization was performed using the ChemiDoc XRS+ (Bio-Rad).

### 2.8. DCF-DA Staining

To estimate the cellular reactive oxygen species, we used the DCF-DA assay (Invitrogen). HF cells were cultured at a density of 1 × 10^5^ cells/mL in a 96-well black plate for a duration of 24 h. Following this incubation, a serum-free medium was introduced to each well to facilitate cell starvation, which lasted for 3 h. Oxidative stress was induced using 8.8 mM of H_2_O_2_, a known ROS. After the starvation period, 100 μL of 10 μM DCF-DA was added to each well and the cells were stained for 30 min. Subsequently, a co-treatment of 8.8 mM of H_2_O_2_ and QGASE at specified concentrations in a serum-free medium was administered and the cells were incubated for an additional hour. Fluorescent cells were then visualized using an inverted microscope, and fluorescence intensity was quantified at excitation and emission wavelengths of 475 nm and 535 nm, respectively, utilizing a spectrophotometer (Molecular Devices, Sunnyvale, CA, USA; SpectraMax iD3).

### 2.9. Ethics for Animal Tests

This study was conducted in accordance with the guidelines set forth by the Standard Protocol for Animal Studies at the Department of Laboratory Animal Resources, Yonsei Hospital Biomedical Research Institute. The experimental protocol was approved by the Institutional Animal Care and Use Committee of our institute (Permit No. 2021-0184). All necessary measures were implemented in this study to minimize suffering, stress, discomfort, and the number of animals utilized.

### 2.10. Mouse Skin Wound Model

Six-week-old male ICR mice were procured from ORIENT BIO (Seongnam, Republic of Korea) and allowed a one-week acclimatization period. The breeding room was maintained under controlled conditions, featuring a 12 h light/dark cycle, a temperature range of 22 °C to 24 °C, and humidity levels between 50% and 60%. During the experimental period, the mice were provided with standard feed and water ad libitum. The fur in the relevant areas of both the experimental and control groups was removed using an electric clipper and a depilatory agent. An epidermal and dermal wound was created using an 8 mm biopsy punch. Four circular wounds, each measuring 8 mm in diameter, were inflicted on two distinct skin sites, positioned 2 cm from the vertebra, utilizing the 8 mm biopsy punch. QGASE was mixed with 10% sodium carboxymethyl cellulose (CMC) at doses of 1, 10, and 40 mg/g. Once daily for 9 days, test material (100 μL) was administered to the wound region until complete epithelialization. Digital cameras were used to capture images of the cutaneous wounds in each group at 0, 3, 6, and 9 days after the test substance was applied to track the wound-healing process. Every three days, the wound size was measured using a digital caliper.

### 2.11. Histopathological Staining of Wound Area

The skin tissue surrounding the wound was preserved in a 10% neutral formalin solution for a duration of 24 h following the conclusion of the experiment. The material was subsequently subjected to the standard procedures of water charging, dehydration, hyalinization, and permeation prior to being embedded in paraffin [24]. The embedded tissue was subsequently sectioned into 4 μm thick slices and stained with hematoxylin and eosin (H&E) to assess alterations in the dermal tissue, including increases in epidermal thickness and the presence of inflammatory cell infiltration. Furthermore, the embedded tissue underwent Masson’s trichrome staining to evaluate changes in dermal collagen, utilizing optical microscopy (Leica, Wetzlar, Germany). Each region was assigned a score ranging from 0 to 4, which was determined by evaluating the presence of inflammatory cells, the extent of collagen deposition, the degree of angiogenesis, the formation of granulation tissue, and the process of re-epithelialization. Each feature was assessed using a semi-quantitative scale ranging from 0 = absent to 4 = prominent) based on clearly defined and reproducible histological characteristics, as outlined by Zaini (Table 2) [25].

### 2.12. Liquid Chromatography—Mass Spectrometry (LC–MS/MS)

The QGASE was dissolved in each solvent to a concentration of 30 mg/mL and subsequently filtered through a 0.45 μm filter unit prior to high-performance liquid chromatography (HPLC) analysis. A quantitative analysis of the phenolic compound extracts was performed using liquid chromatography–tandem mass spectrometry (LC-MS/MS) with a Nexera X2 system (Shimadzu Co., Kyoto, Japan). The analysis utilized a C18 column (Quasar 50 × 2.1 mm, 1.7 μm, PerkinElmer Co., Shelton, CT, USA). The mobile phase consisted of water with 0.1% formic acid (referred to as A) and acetonitrile with 0.1% formic acid (referred to as B). The gradient conditions for the mobile phase were established as follows: at 0.5 min, B was set to 5%; at 1 min, B remained at 5%; at 10 min, B increased to 95%; and at 18 min, B was maintained at 95%. The sample injection volume was 5 μL and the analysis was conducted at a flow rate of 0.3 mL/min. The standard materials utilized in this study included protocatechuic acid (CFN97568), catechin (CFN99646), ellagic acid (CFN98716), epicatechin (CFN98781), epicatechin gallate (CFN98570), gallic acid (CFN99624), isoquercitrin (CFN98753), kaempferol–3–O-(2′-6′-di-O-trans-p-coumaroyl)-beta-D-glucopyranoside (CFN92386), myricitrin (CFN99840), quercetin (CFN99272), rutin (CFN99642), and tiliroside (CFN98026), all of which were procured from Sigma-Aldrich (St. Louis, MO, USA).

### 2.13. Statistical Analysis

All quantitative values are expressed as the mean ± standard deviation. Each experiment comprised three replicates for each experimental variable and was conducted between three and five times. Figures illustrate the results of representative experiments. Statistical differences were assessed using Student’s *t*-test, with significance established at *p* < 0.05.

## 3. Results

### 3.1. QGASE Repaired the Wound Area in Human Dermal Fibroblast (HF) Cells

To investigate the wound-healing effects of QGASE, we examined its effects on the migration of HF cells. As shown in Figure 1A, QGASE facilitated the closure of the wound area within the cells in a concentration-dependent manner. The analysis of the percentage of the wound-healing area indicated that treatment with the compound resulted in recovery in the wound region, in contrast to the vehicle control, also in a dose-dependent manner (Figure 1B). However, these effects were not cytotoxic or proliferative at the concentrations tested (Figure 1C, Appendix A). These results suggest that QGASE had wound-healing effects without exerting cytotoxicity and proliferation in HF cells.

### 3.2. QGASE Increased the Levels of Wound-Healing Biomarkers in HF Cells

To identify how QGASE affected wound healing, we investigated the levels of wound-healing biomarkers. As shown in Figure 2A, QGASE enhanced the transcriptional expression of COL1A1, COL3A1, and VEGF-A in a dose-dependent manner. Protein analysis showed that QGASE increased the expression of wound-healing markers, such as β-catenin, fibronectin, endothelin, α-SMA, keratin14, COL1A1, and vimentin compared with the vehicle control (Figure 2B). These results suggested that QGASE upregulated the mRNA and protein levels of biomarkers associated with wound healing. Thus, we confirmed that QGASE exhibited wound-healing activities in HF cells.

### 3.3. QGASE Reduced ROS Levels in HF Cells

To analyze the anti-inflammatory activity of QGASE, we induced ROS in HF cells through H_2_O_2_ exposure. DCF-DA staining showed that H_2_O_2_ increased intracellular ROS levels, but QGASE treatment attenuated the intracellular ROS levels in HF cells in a concentration-dependent manner (Figure 3A). The fluorescence of H_2_O_2_-induced ROS was decreased by QGASE treatment in a concentration-dependent manner (Figure 3B). To evaluate that this ROS-scavenging effect was not cell cytotoxicity, we performed a CCK-8 assay under the same conditions. As shown in Figure 3C (Appendix A), the antioxidant activity of QGASE was not cytotoxic at the tested doses. Thus, QGASE exhibited a ROS-scavenging effect without any cytotoxicity in HF cells.

### 3.4. QGASE Recovered H_2_O_2_-Enhanced Wounds in HF Cells

To analyze the migratory effect of QGASE on H_2_O_2_-induced wounds, we stimulated the HF cells with H_2_O_2_ to mimic wounds in vitro. As shown in Figure 4A, the wounds induced by H_2_O_2_ were filled with cells after QGASE treatment in a concentration-dependent manner. As shown in Figure 4B, the recovery percentage of the wound area was low when the HF cells were treated with H_2_O_2_, whereas QGASE treatment resulted in recovery of the wound site (Figure 4B). These effects were not due to cytotoxicity or cell proliferation (Figure 4C; Appendix A). To determine how QGASE induced a wound recovery effect, we performed qRT-PCR of wound-healing markers such as vimentin, fibronectin, and COL3A1. As shown in Figure 4D, the attenuated expression of the markers caused by H_2_O_2_ was increased by QGASE treatment. Based on these results, we confirmed that QGASE showed H_2_O_2_-enhanced wound-healing activity by increasing the expression of markers involved in wound healing.

### 3.5. QGASE Showed Wound-Healing Effects In Vivo

To clarify the wound-healing effects of QGASE, we performed an in vivo assay using a mouse wound model. As shown in Figure 5A, QGASE repaired wounds in a dose-dependent manner. The percentage of the wound area exhibited a dose-dependent effect on healing (Figure 5B left panel). The wound area exhibited a concentration-dependent effect on wound healing (Figure 5B right panel). Furthermore, QGASE suppressed the expression of inflammatory cytokines in a concentration-dependent manner (Figure 5C). We confirmed by H&E staining that QGASE restored the wound area (Figure 5D). The staining revealed the epithelialization, hemorrhage, inflammation, granulation tissue formation, and neovascularization of the tissue. The skin condition scores improved in a dose-dependent manner (Table 3). In addition, Masson’s trichrome staining revealed that collagen deposition was increased by QGASE treatment (Figure 5E; Appendix A; Table 3). Through these results, we confirmed that QGASE had skin wound-healing and anti-inflammatory effects in vivo.

### 3.6. QGASE Contained Various Polyphenolic Compounds

To determine how many polyphenolic compounds are present in *Quercus glauca* Thunb. acorn seed coat water extract (QGASE), the concentrations of 12 compounds were determined using liquid chromatography—mass spectrometry (LC–MS/MS). A total of twelve standard polyphenolic compounds were studied to quantify the concentration of these compounds in the extract (Appendix A). The results showed that gallic acid, protocatechuic acid, catechin, epicatechin, and rutin had the highest amounts (Table 4).

## 4. Discussion

The objective of our research was to investigate the potential role of QGASE in skin injury. Our findings indicate that QGASE effectively enhances wound closure in HF cells without causing cytotoxicity or promoting cell proliferation. To elucidate the underlying mechanisms, we investigated the expression of critical wound-healing markers.

Wound healing is a multifaceted phenomenon distinguished by a progression of sequential stages: hemostasis, inflammation, angiogenesis, growth, re-epithelialization, and remodeling [26]. These phases involve a coordinated interplay of various factors aimed at restoring tissue integrity. For example, VEGF facilitates processes such as angiogenesis, collagen synthesis, and epithelialization [27]. β-catenin assumes a crucial role throughout the healing process by bridging early events, such as TGF-β signaling, with later stages, including MMP activation [28]. The Wnt/β-catenin pathway is indispensable for fibroblast functions like migration, proliferation, and differentiation, thereby influencing both regenerative processes and fibrotic responses [29,30].

Fibronectin plays a pivotal role, particularly in the initial stages of wound healing. It interacts with platelets and fibrin to enhance clot stability, promoting cell adhesion and migration [14]. Later, it facilitates epidermal keratinocyte migration across the wound bed [31]. The assembly of fibronectin involves contributions from platelets and cells, with plasma-derived fibronectin being more prevalent in the early stages and cell-derived fibronectin becoming dominant in the later stages to support tissue remodeling [32]. TGF-β stimulates the differentiation of fibroblasts into myofibroblasts, a process identified by the expression of α-SMA, which performs a critical part in wound contraction and matrix restructuring [33,34], The upregulation of α-SMA expression is a distinguishing feature of mature myofibroblasts and is prominently observed during the wound-healing process [35,36].

Re-epithelialization encompasses the processes of keratinocyte migration and proliferation, with keratin 14 serving as a prominent marker [37]. Collagen, specifically types I and III, offers structural reinforcement and acts as a framework for cellular interactions [38,39]. Vimentin, a cytoskeletal protein, experiences upregulation during the wound-healing process and contributes to cell migration and signaling [40,41]. Our results suggest that QGASE enhances the expression of key markers associated with wound healing in a concentration-dependent manner throughout the stages of wound repair. This indicates that QGASE may promote wound healing by modulating essential signaling pathways and cellular processes.

Exorbitant production of ROS has been shown to be involved in inflammatory responses and compromised wound healing [42]. PMNs are a major origin source of ROS, which can lead to endothelial dysfunction and tissue damage at the site of inflammation [43]. Considering the significant impact of oxidative stress on wound healing, this study explores the antioxidant properties of QGASE. Our findings indicate that QGASE efficiently scavenges ROS induced by H_2_O_2_ in a manner that is dependent on the dosage, while maintaining cell viability. To estimate the potential therapeutic impacts of QGASE in the context of wound healing, we utilized an H_2_O_2_-induced wound model that replicates the delayed healing process linked to inflammation. QGASE notably promoted cell migration and wound closure, even under conditions of oxidative stress, suggesting that its capacity to improve wound healing is caused by the inhibition of inflammatory conditions. ROS have been implicated in a spectrum of cellular dysfunctions, encompassing oxidative damage to lipids, proteins, and nucleic acids [44]. Moreover, inflammation is exacerbated by oxidative stress induced by H_2_O_2_, which substantially contributes to the progression of chronic wounds [45]. Our observations align with these discoveries, as we noted a reduction in the expression of markers associated with wound healing following H_2_O_2_ treatment, indicating its adverse effects on the healing process. Significantly, treatment with QGASE effectively reversed the inhibitory effects of H_2_O_2_ on the transcriptional activity of genes related to wound healing. Collectively, the findings indicate that QGASE exhibits strong antioxidant and anti-inflammatory characteristics, which play a role in its effectiveness in enhancing the process of wound healing.

To assess the medicinal potential of QGASE in vivo, a murine wound model was utilized. The administration of QGASE led to enhanced wound closure in a manner dependent on the dosage. Excessive inflammation, characterized by increased levels of pro-inflammatory cytokines such as IL-1β, IL-6, and TNF-α, is recognized to hinder wound healing and facilitate scar formation [46,47]. The findings from our study suggest that QGASE not only enhances the speed of wound healing but also significantly decreases the levels of these inflammatory markers at the sites of skin wounds. Histopathological examination further validated the positive impacts of QGASE on wound healing, as demonstrated by enhanced wound-healing parameters. Collectively, these findings suggest that QGASE shows promising therapeutic potential in expediting wound repair and alleviating inflammation.

To identify the bioactive compounds responsible for QGASE’s wound-healing effects, comprehensive polyphenol profiling was conducted using LC–MS/MS. Gallic acid, protocatechuic acid, catechin, epicatechin, and rutin were identified as the predominant polyphenols. Previous studies have reported the antioxidant and various health-promoting properties of polyphenols, including gallic acid [48,49,50]. Notably, gallic acid has been specifically linked to wound healing [51]. Protocatechuic acid, a polyphenolic compound, exhibits antioxidant properties as well as wound-healing effects [52]. Additionally, catechin and epicatechin are natural polyphenolic compounds that possess anti-inflammatory, antioxidant, and free radical scavenging properties [53]. These findings suggest that the observed wound-healing effects of QGASE may be attributed, in part, to its polyphenolic constituents. Further investigations are warranted to elucidate the specific contributions of these compounds to the overall efficacy of QGASE.

## 5. Conclusions

This study demonstrates the potential therapeutic effectiveness of QGASE in wound healing. The findings suggest that QGASE promotes wound closure in both in vitro and in vivo environments by stimulating HF cell migration, exhibiting potent antioxidant and anti-inflammatory properties. By reducing inflammatory cytokines and enhancing the expression of wound-healing markers, QGASE significantly improves the wound-healing process. Further investigation is required to determine the specific molecular targets and bioactive compounds responsible for these effects. Nevertheless, the results indicate that QGASE shows potential as a candidate for the development of novel wound-healing therapies.

## Figures and Tables

**Figure 1 biology-13-00775-f001:**
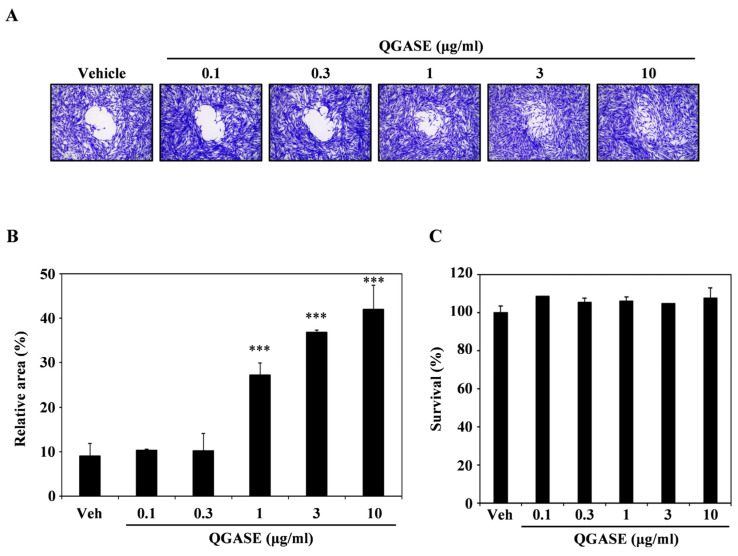
*Quercus glauca* Thunb. acorn seed coat water extract (QGASE) recovered the wound region in human dermal fibroblast cells (HF cells). (**A**) HF cells were cultured on a migration plate for 24 h, and then the treatment of QGASE was applied in serum-free media. After an additional 24 h, the cells were fixed and stained using a cell staining solution. (**B**) The relative area was measured using Image J program. *** *p* < 0.001 (versus vehicle control). (**C**) The effect of QGASE on the viability of HF cells was evaluated with the CCK-8 assay. Data are expressed as the mean ± SD (n = 3).

**Figure 2 biology-13-00775-f002:**
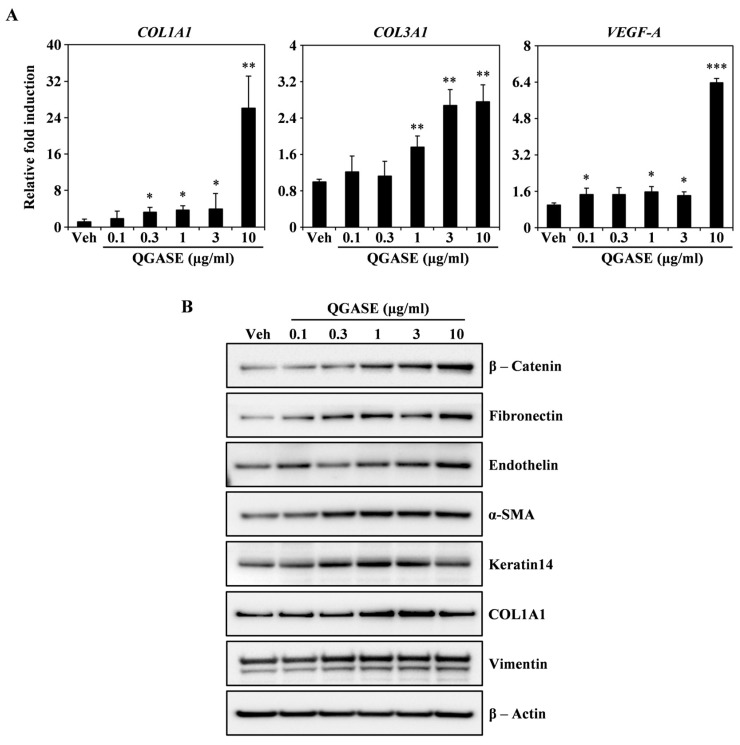
QGASE increased the expression of wound-healing molecules involved in HF cells. (**A**) mRNA expression levels in HF cells stimulated with vehicle or QGASE (0.1, 0.3, 1, 3, and 10 μg/mL) measured using qRT-PCR. *Actin* and *HPRT1* were used as the internal controls; * *p* < 0.05; ** *p* < 0.01; *** *p* < 0.001 (versus control). Data are expressed as the mean ± SD (n = 3). (**B**) The effect of QGASE on the protein expression levels of migration molecules evaluated using western blot analysis. β-actin was used as the internal control. The result is representative of three independent experiments yielding similar results.

**Figure 3 biology-13-00775-f003:**
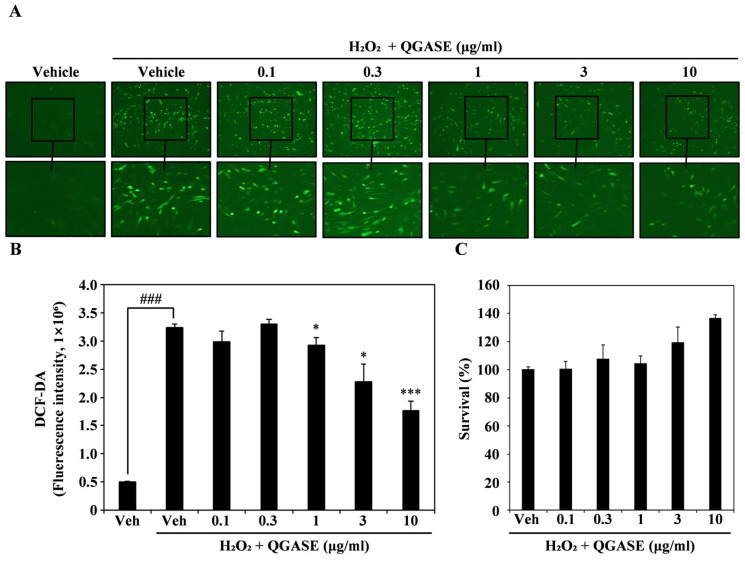
QGASE decreased ROS levels in HF cells. (**A**) HF cells were starved for 3 h and pretreated with 10 μM DCF-DA and QGASE for 1 h. Then, cell morphology was observed under a fluorescence microscope. (**B**) DCF-DA fluorescence intensity determined using a fluorescence spectrophotometer at 475–535 nm. ### *p* < 0.001 (versus vehicle control); * *p* < 0.05; *** *p* < 0.001 (versus the H_2_O_2_-treated group). (**C**) Effect of QGASE on HF viability evaluated using the CCK-8 assay. Data are expressed as the mean ± SD (n = 3).

**Figure 4 biology-13-00775-f004:**
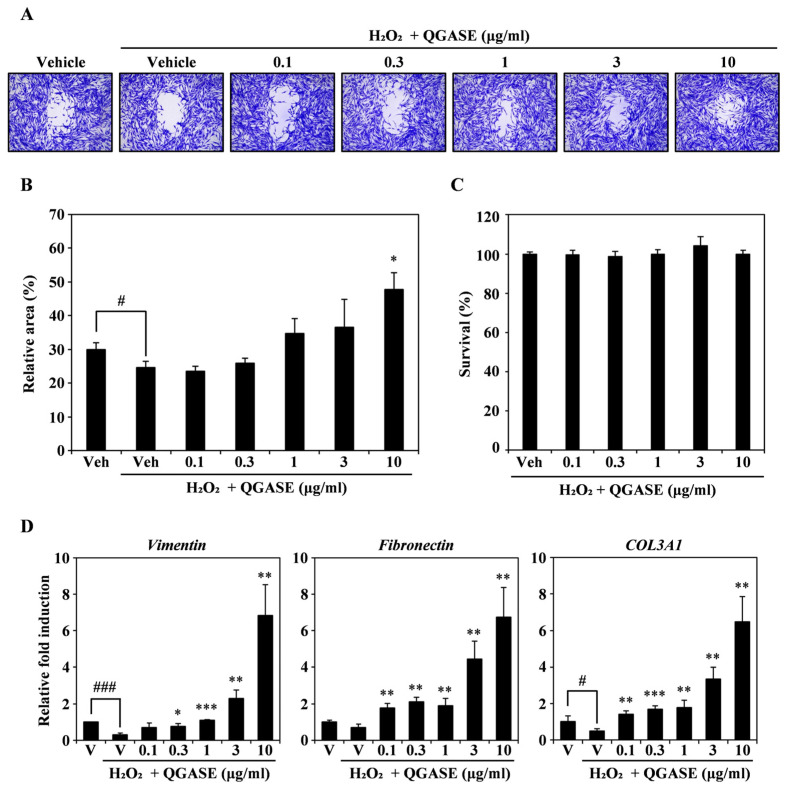
QGASE repaired an H_2_O_2_-enhanced wound in HF cells. (**A**) HF cells were cultured with vehicle or QGASE (0.1, 0.3, 1, 3, and 10 μg/mL) in the presence of H_2_O_2_ (500 μM) on a migration plate. After wound healing, cell fixation and staining were performed. (**B**) The wound was measured using the Image J program. # *p* < 0.05 (versus control); * *p* < 0.05 (versus the H_2_O_2_-treated group). (**C**) QGASE cytotoxicity was evaluated using the CCK-8 assay. (**D**) mRNA expression levels (measured using qRT-PCR) in HF cells stimulated with vehicle control or QGASE (0.1, 0.3, 1, 3, and 10 μg/mL) in the presence of H_2_O_2_ (500 μM). Actin was used as the internal control. The reaction was performed in triplicate. # *p* < 0.05; ### *p* < 0.001 (versus control); * *p* < 0.05; ** *p* < 0.01; *** *p* < 0.001 (versus the H_2_O_2_-treated group). Data are expressed as the mean ± SD (n = 3).

**Figure 5 biology-13-00775-f005:**
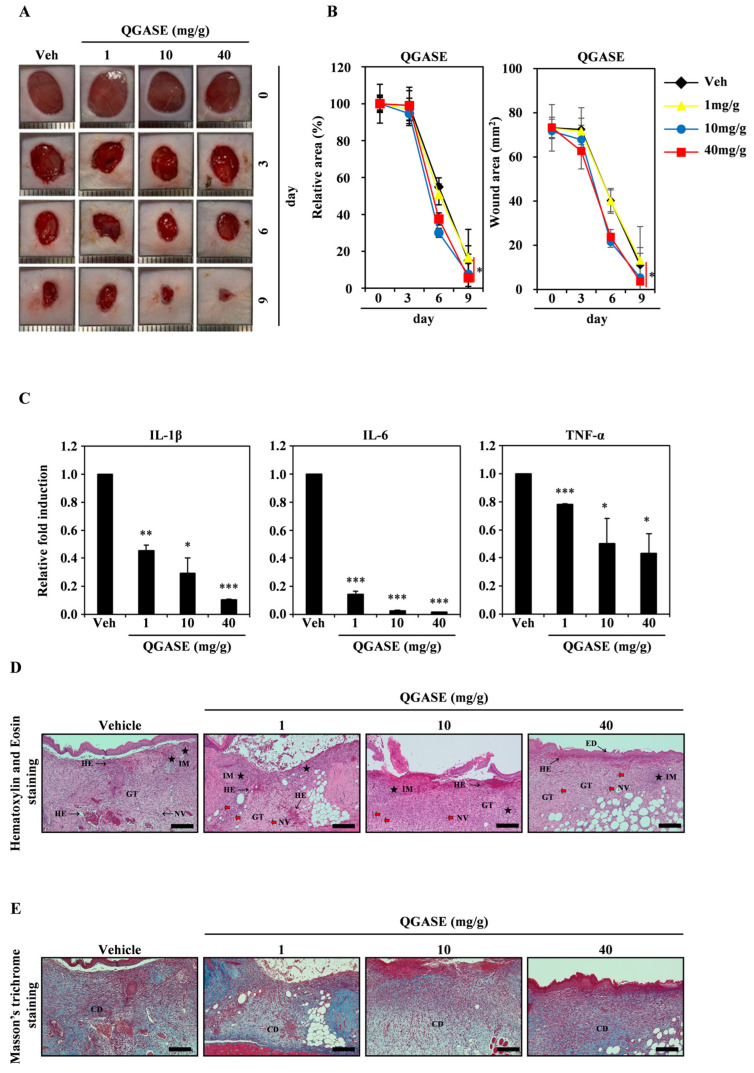
QGASE showed wound-healing effects in vivo. (**A**) A full-thickness skin excision of 8 mm in diameter was performed on the back of each mouse. QGASE was applied to the wounded area for 9 days. (**B**) The area was measured utilizing a digital caliper. * *p* < 0.05 (versus vehicle control). Data are expressed as the mean ± SD (n = 3). (**C**) The mRNA expression levels were analyzed using qRT-PCR. * *p* < 0.05, ** *p* < 0.01, *** *p* < 0.001 (versus vehicle control). Data are expressed as the mean ± SD (n = 3). (**D**) The skin tissue was fixed in 3.7% formalin for 24 h, then sliced to a thickness of 4 μm. Then, it was analyzed using H&E staining (**E**) and Masson’s trichrome staining (n = 3). GT—granulation tissue formation; HE—hemorrhage, Thick Black Arrow symbol; IM—inflammation, Black Star symbol; NV—neovascularization, Red Arrow symbol; ED—epithelialization, Thin Black Arrow symbol; CD—collagen deposition. scale bar is 200 μm.

**Table 1 biology-13-00775-t001:** Primer sequences used in this study.

Target Gene	Forward Primer (5′–3′)	Reverse Primer (5′–3′)
*COL1A1*	CCGTGCCCTGCCAGATC	CAGTTCTTGATTTCGTCGCAGATC
*COL3A1*	TGGAGGATGGTTGCACGAAA	AAAAGCAAACAGGGCCAACG
*VEGF-A*	ATAAGTCCTGGAGCGTTCCCT	GGCAGCGTGGTTTCTGTATC
*Vimentin*	AACTTAGGGGCGCTCTTGTC	TGAGGGCTCCTAGCGGTTTA
*Fibronectin*	ACAAGCATGTCTCTCTGCCA	TTTGCATCTTGGTTGGCTGC
*TNF-α*	TAGCCCACGTCGTAGCAAAC	CTCAAAGTAGACCTGCCC
*IL-1β*	TGCCACCTTTTGACAGTGATG	AAGGTCCACGGGAAAGACAC
*IL-6*	CAACGATGATGCACTTGCAGA	TGGAAATTGGGGTAGGAAGGAC
*HPRT1*	GACCAGTCAACAGGGGACAT	GCTTGCGACCTTGACCATCT
*β-actin*	AAGGATTCCTATGTGGGCGAC	CGTACAGGGATAGCACAGCC

**Table 2 biology-13-00775-t002:** Scoring system for histological change in skin wound healing.

Score	0	1	2	3	4
Epithelialization	Absence of epithelial proliferation in 70% of tissue	Poor epidermal organization in 60% of tissue	Incomplete epidermal organization in 40% of tissue	Moderate epithelial proliferation in 60% of tissue	Complete epidermal remodeling in 80% of tissue
Hemorrhage	Absence of hemorrhage	1–2 per site of hemorrhage	3–4 per site of hemorrhage	5–6 per site of hemorrhage	>7 per site of hemorrhage
Granulation tissue formation	Immature and inflammatory tissue in 70% of tissue	Thin, immature, and inflammatory tissue in 60% of tissue	Moderate remodeling in 40% of tissue	Thick granulation layer in 60% of tissue	Complete tissue organization in 80% of tissue
Neovascularization	Absence of angiogenesis	1–2 vessels per site	3–4 vessels per site	5–6 vessels per site	>7 vessels per site
Collagen deposition	Absence of collagen deposition	Focal presence in fibroblasts around new capillaries	Moderate amount in the repair tissue	Dominant feature	
Inflammation	Inflammatory cell infiltration was scored as follows: - mild - moderate - severe				

**Table 3 biology-13-00775-t003:** Histological analysis of wound healing using H&E staining and Masson’s trichrome staining with QGASE.

Group Dose	Vehicle	QGASE		
	0	1	10	40
Score				
Skin				
– Epithelialization	0	0	0	4
– Hemorrhage	3	3	2	1
– Inflammation	3	3	2	1
– Granulation tissue formation	2	4	2	4
– Neovascularization	1	4	2	3
Dermis				
– Collagen deposition	1	1	2	3

The wound area was quantified after H&E staining and Masson’s trichrome staining (n = 12).

**Table 4 biology-13-00775-t004:** Concentrations of the polyphenolic compounds in QGASE.

Identification	Formula	Molar Mass	mg/kg
Epicatechin gallate	C_22_H_18_O_10_	442.37 g/mol	3.00 ± 0.56
Protocatechuic acid	C_7_H_6_O_4_	154.12 g/mol	573.33 ± 56.96
Ellagic acid	C_14_H_6_O_8_	302.197 g/mol	8.37 ± 1.40
Gallic acid	C_7_H_6_O_5_	170.12 g/mol	1407.78 ± 128.51
Isoquercitrin	C_21_H_20_O_12_	464.0955 g/mol	0.00 ± 0.00
Kaempferol-3-O-(2′6′-di-O-trans-p-coumaroyl)-β-D-glucopyranoside	C_39_H_32_O_15_	740.7 g/mol	0.00 ± 0.00
Myricitrin	C_21_H_20_O_12_	464.37 g/mol	0.00 ± 0.00
Tiliroside	C_30_H_26_O_13_	594.5 g/mol	0.52 ± 0.23
Catechin	C_15_H_14_O_6_	290.26 g/mol	143.67 ± 17.13
Epicatechin	C_15_H_14_O_6_	290.27 g/mol	83.33 ± 4.58
Quercetin	C_15_H_10_O_7_	302.236 g/mol	7.89 ± 1.54
Rutin	C_27_H_30_O_16_	610.517 g/mol	63.44 ± 10.01

## Data Availability

The data presented in this study are available on request from the corresponding author.

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
