# Peer review of "Quercus glauca Acorn Seed Coat Extract Promotes Wound Re-Epithelialization by Facilitating Fibroblast Migration and Inhibiting Dermal Inflammation"

_biology, 2024, doi:10.3390/biology13100775_

Round 1
Reviewer 1 Report
Comments and Suggestions for Authors
The authors generated an aqueous solution from the acorn seed coat of Quercus glauca (QGASE) and tested its potential to promote skin wound healing.
For this, they used various cellular, molecular, and biochemical methods in vitro, a wound closure assay mouse model in vivo, and finally analyzed QGASE for the presence of polyphenolic compounds that could potentially be contributing to the observed effects.
Unfortunately, there is no systematic analysis of whether QGASE is more potent than single polyphenolic compounds, but such an analysis would go far beyond the scope of this study.
The authors found that QGASE has antioxidant effects, attenuates oxidative stress, and promotes wound healing in vitro and in vivo. The in vitro migration assay was performed using a human fibroblast cell line. The wound closure by fibroblasts is a rather late event in the wound healing process, while inflammatory processes are more immediate. In addition, also keratinocyte proliferation and migration contribute to the wound-healing process. It would have been interesting to address QGASE activity on the proliferation and migration properties of keratinocytes. However, this can also be considered to be outside the scope of this study.
The expression of a few inflammatory cytokines in response to QGASE was analyzed by qPCR, but the display of the results is somehow misleading (for details, see my comments below).
Overall, the experiments appear to be thoroughly performed, technically sound, and scientifically accurate. Apart from a few exceptions, the description of the results is good.
There are only a few, but critical concerns that need to be clarified.
1) Migration assays.
Filling the empty space can be achieved by migration of cells (without an increase in cell number) due to the migratory capacity of the cells or when an increased cell number of proliferating cells needs to occupy the empty space. Specific stimuli may influence either the migratory or the proliferative capacity of cells or both. The migration assay used here cannot distinguish between both effects. If the authors want to specify this, I recommend performing migration assays under cell cycle arrest using, for example, serum-free conditions, inhibitors (e.g. mitomycin), or gamma irradiation.
In this respect, the cell counting kit-8 that was used in this study to assess cytotoxic effects can provide some evidence. Figure 1C shows survival rates of more than 100% under different conditions of QGASE treatment, which may indicate a positive effect on the proliferation rate of the HF cell line.
a) The authors should clearly state the baseline condition to which the survival rates under QGASE treatment refer (HF cells with vehicle only?).
b) The same applies to the data in Figure 3C.
c) Providing the CCK-8 absorbance data at 450 nm would not only show that QGASE is not cytotoxic but also whether or not it has a dose-dependent effect on the proliferation of HF cells.
2) Mouse skin wound model
In the murine skin wound model, an 8 mm biopsy punch was used to create the skin wounds. As described in the Materials and Methods section, the wounds were photographed every three days using a digital camera and measured with a digital caliper. The quantification of wound area closure is depicted in Figure 5B.
a) It seems that the y-axis of Figure 5B is missing a label. I assume it should represent the percent of the initial wound area.
b) The wounds and their closure areas are irregular in shape. Instead of, or in addition to, using a caliper to calculate the wound area, the authors could have utilized imageJ, similar to the migration assays, to measure the wound size within a specified field of view. The data for wound areas could then be presented in mm^2.
c) Additionally, the legend states a significant difference of *p < 0.05 (versus vehicle control), but no asterisks are shown in the plot.
d) Page 5, line 209, the authors wrote: “A digital caliper was used to gauge the wound size for 3 days.” Is this correct, or did they mean: “The wound size was measured every 3 days using a digital caliper.”?
3) Histological examination of wound areas in the mouse skin wound model
The description in the Materials and Methods section doesn’t correspond to the results shown.
a) Page 5, line 215: “…embedded tissue was subsequently sliced into 4 mm-thick pieces…” should be “…4 µm-thick…” (correct in the figure legend).
b) The section states that “…the embedded tissue was stained with toluidine blue to determine the location of mast cells or with Masson's trichrome…”. While Masson's trichrome staining is shown in Figure 5E and scored in Table 2, there are no results of a toluidine blue staining and mast cells shown.
c) The method section states further that “Each region received a score between 0 and 3 based on the number of inflammatory cells, collagen deposition, angiogenesis, creation of granulation tissue, and re-epithelialization.” However, Table 2 shows scores up to 4. Please clarify. In addition, Figures 5D and 5E show representative images of tissue sections. According to Figure 5B, three mice were wounded. Please indicate how many sections from how many mice were stained and analyzed. How many sections were analyzed for the scoring shown in Table 2? Are the scoring grades in Table 2 the mean values or kind of a summarized result? The results would be more convincing if they were not only based on n = 1 section staining and examination.
d) Representative sections from all three mice can be included as supplemental material.
4) The mRNA expression levels of proinflammatory cytokines
The influence of QGASE on the mRNA expression levels of IL-1beta, IL-6, and TNFalpha is shown as part of Figure 5 between the in vivo wound measurements and the histopathological staining of the wound areas. This is misleading since, according to the Materials and Methods section, the mRNA isolation and qPCR were described only for HF cells treated with H2O2. Therefore, the cytokine expression results either belong to the results shown in Figure 4, or it should be described in more detail how the data were generated (e.g. was the mRNA isolated from the skin, at what time point of the in vivo experiment, etc.) if the expression was assessed in the in vivo wound healing experiment.
Minor points:
5) Figure 3B: The label on the y-axis indicates “fluorescence intensity, 1 x 10^6,” but it then ranges from 500000 to 4000000. I assume it should read 0.5 to 4. Please check and correct this accordingly.
6) Page 4, line 167: “anti-α-smooth muscle…”, page 7, line 269: “…α-smooth muscle…”, and the label in Figure 2 B: “α-smooth muscle” needs to be corrected to “α-smooth muscle actin”. This term is usually abbreviated as “α-SMA” (see also the Discussion section, page 14, line 379), which could be used for labeling the western blot result.
7) The layout of Table 2 should be refined (i.e. the “groups” are below the “dose” column).
Comments on the Quality of English LanguageThe authors should thoroughly check the text again for grammatical and spelling errors. It’s not a lot and mainly hyphenation.
For example:
Page 2, line 68:
“…chemoattract-ants for inflammatory…”
Page 13, lines 351-352:
“The peak was exhibited 12 standard compounds at 100 ppb.”
Page 14, line 385:
“…and acts as a frame-work for cellular…”
Author Response
â–ºReviewer 1’s comments:
The authors generated an aqueous solution from the acorn seed coat of Quercus glauca (QGASE) and tested its potential to promote skin wound healing.
For this, they used various cellular, molecular, and biochemical methods in vitro, a wound closure assay mouse model in vivo, and finally analyzed QGASE for the presence of polyphenolic compounds that could potentially be contributing to the observed effects.
Unfortunately, there is no systematic analysis of whether QGASE is more potent than single polyphenolic compounds, but such an analysis would go far beyond the scope of this study.
The authors found that QGASE has antioxidant effects, attenuates oxidative stress, and promotes wound healing in vitro and in vivo. The in vitro migration assay was performed using a human fibroblast cell line. The wound closure by fibroblasts is a rather late event in the wound healing process, while inflammatory processes are more immediate. In addition, also keratinocyte proliferation and migration contribute to the wound-healing process. It would have been interesting to address QGASE activity on the proliferation and migration properties of keratinocytes. However, this can also be considered to be outside the scope of this study.
The expression of a few inflammatory cytokines in response to QGASE was analyzed by qPCR, but the display of the results is somehow misleading (for details, see my comments below).
Overall, the experiments appear to be thoroughly performed, technically sound, and scientifically accurate. Apart from a few exceptions, the description of the results is good.
There are only a few, but critical concerns that need to be clarified.
1) Migration assays.
Filling the empty space can be achieved by migration of cells (without an increase in cell number) due to the migratory capacity of the cells or when an increased cell number of proliferating cells needs to occupy the empty space. Specific stimuli may influence either the migratory or the proliferative capacity of cells or both. The migration assay used here cannot distinguish between both effects. If the authors want to specify this, I recommend performing migration assays under cell cycle arrest using, for example, serum-free conditions, inhibitors (e.g. mitomycin), or gamma irradiation.
In this respect, the cell counting kit-8 that was used in this study to assess cytotoxic effects can provide some evidence. Figure 1C shows survival rates of more than 100% under different conditions of QGASE treatment, which may indicate a positive effect on the proliferation rate of the HF cell line.
- A) Thank you for your thorough review. We conducted the migration assay and cell proliferation assay illustrated in Figure 1 under serum free conditions. To provide further clarification, we have included this information into the figure legend.
- The authors should clearly state the baseline condition to which the survival rates under QGASE treatment refer (HF cells with vehicle only?).
- A) Thank you for your insightful suggestion. In accordance with your recommendation, we have incorporated the vehicle group into Figure 1C to enhance the clarity of the baseline condition.
- The same applies to the data in Figure 3C.
- A) Thank you for your suggestion. As recommended, we added vehicle group into Figures 3C and 4C for clarifying the baseline condition.
- Providing the CCK-8 absorbance data at 450 nm would not only show that QGASE is not cytotoxic but also whether or not it has a dose-dependent effect on the proliferation of HF cells.
- A) Thank you for this valuable suggestion. As recommended, we have included the CCK-8 absorbance data at 450 nm in Supplementary Figure 1c.
2) Mouse skin wound model
In the murine skin wound model, an 8 mm biopsy punch was used to create the skin wounds. As described in the Materials and Methods section, the wounds were photographed every three days using a digital camera and measured with a digital caliper. The quantification of wound area closure is depicted in Figure 5B.
- a) It seems that the y-axis of Figure 5B is missing a label. I assume it should represent the percent of the initial wound area.
- A) We appreciate your insightful feedback and confirm that your observations are accurate. In response, we have revised the y-axis label in Figure 5B to reflect the percentage of the initial wound area.
- b) The wounds and their closure areas are irregular in shape. Instead of, or in addition to, using a caliper to calculate the wound area, the authors could have utilized imageJ, similar to the migration assays, to measure the wound size within a specified field of view. The data for wound areas could then be presented in mm^2.
- A) Thank you for your thorough review. In accordance with your suggestion, we have utilized the ImageJ program to calculate the wound surface area. Subsequently, we have included a graph in Figure 5B (right graph) that illustrates the calculated average value and standard deviation.
- c) Additionally, the legend states a significant difference of *p < 0.05 (versus vehicle control), but no asterisks are shown in the plot.
- A) Thank you for your careful review. As per your suggestion, we have marked an asterisk in the graph.
- d) Page 5, line 209, the authors wrote: “A digital caliper was used to gauge the wound size for 3 days.” Is this correct, or did they mean: “The wound size was measured every 3 days using a digital caliper.”?
- A) We appreciate your thorough review and acknowledge your correctness. Consequently, we have revised the sentence to read, three days, we measured the wound size using a digital caliper.
3) Histological examination of wound areas in the mouse skin wound model
The description in the Materials and Methods section doesn’t correspond to the results shown.
- a) Page 5, line 215: “…embedded tissue was subsequently sliced into 4 mm-thick pieces…” should be “…4 µm-thick…” (correct in the figure legend).
- A) We acknowledge that there was an error in the unit of measurement, and we appreciate your thorough review. The unit of measurement has been corrected to 4 micrometers.
- b) The section states that “…the embedded tissue was stained with toluidine blue to determine the location of mast cells or with Masson's trichrome…”. While Masson's trichrome staining is shown in Figure 5E and scored in Table 2, there are no results of a toluidine blue staining and mast cells shown.
- A) We express our gratitude for your thorough review and recognize the inaccuracies in our prior statement. The sentence has been amended to state: the embedded tissue was stained with Masson's trichrome to identify the changes in dermal collagen using optical microscopy
- c) The method section states further that “Each region received a score between 0 and 3 based on the number of inflammatory cells, collagen deposition, angiogenesis, creation of granulation tissue, and re-epithelialization.” However, Table 2 shows scores up to 4. Please clarify. In addition, Figures 5D and 5E show representative images of tissue sections. According to Figure 5B, three mice were wounded. Please indicate how many sections from how many mice were stained and analyzed. How many sections were analyzed for the scoring shown in Table 2? Are the scoring grades in Table 2 the mean values or kind of a summarized result? The results would be more convincing if they were not only based on n = 1 section staining and examination.
- A) Thank you for your insightful comment. We have integrated the skin wound healing scoring guidelines into the Materials and Methods section, as presented in Table 2, and have revised the corresponding sentences accordingly. Furthermore, the figure and table legend specify the number of mice utilized for the sections (n=3) and the number of sections analyzed for scoring (n=12).
- d) Representative sections from all three mice can be included as supplemental material.
- A) We appreciate your valuable suggestion. In accordance with your recommendation, we have included all relevant data pertaining to the histological staining in Supplementary Figure 2.
4) The mRNA expression levels of proinflammatory cytokines
The influence of QGASE on the mRNA expression levels of IL-1beta, IL-6, and TNFalpha is shown as part of Figure 5 between the in vivo wound measurements and the histopathological staining of the wound areas. This is misleading since, according to the Materials and Methods section, the mRNA isolation and qPCR were described only for HF cells treated with H2O2. Therefore, the cytokine expression results either belong to the results shown in Figure 4, or it should be described in more detail how the data were generated (e.g. was the mRNA isolated from the skin, at what time point of the in vivo experiment, etc.) if the expression was assessed in the in vivo wound healing experiment.
- A) Thank you for this important suggestion. In accordance with your recommendation, we have provided a more detailed description of the mRNA isolated from the skin and the qPCR methodology in the Materials and Methods section.
Minor points:
5) Figure 3B: The label on the y-axis indicates “fluorescence intensity, 1 x 10^6,” but it then ranges from 500000 to 4000000. I assume it should read 0.5 to 4. Please check and correct this accordingly.
- A) We acknowledge the error in the unit of measurement and appreciate your thorough review. The measurement range has been corrected to 0.5 to 4.0, 1 x 106.
6) Page 4, line 167: “anti-α-smooth muscle…”, page 7, line 269: “…α-smooth muscle…”, and the label in Figure 2 B: “α-smooth muscle” needs to be corrected to “α-smooth muscle actin”. This term is usually abbreviated as “α-SMA” (see also the Discussion section, page 14, line 379), which could be used for labeling the western blot result.
- A) Thank you for your insightful suggestion. We have corrected the term muscle actin (α-SMA).
7) The layout of Table 2 should be refined (i.e. the “groups” are below the “dose” column).
- A) The columns for group and dose have been integrated and reorganized to improve their clarity and distinguishability.
Comments on the Quality of English Language
The authors should thoroughly check the text again for grammatical and spelling errors. It’s not a lot and mainly hyphenation.
For example:
Page 2, line 68:
“…chemoattract-ants for inflammatory…”
Page 13, lines 351-352:
“The peak was exhibited 12 standard compounds at 100 ppb.”
Page 14, line 385:
“…and acts as a frame-work for cellular…”
- A) The manuscript has undergone a thorough review, and the requisite modifications have been implemented in alignment with the recommendations provided by the reviewers.
We would like to express our sincere gratitude to you and the reviewers for your valuable feedback. We hope that the revised version of our manuscript is now deemed suitable for publication in Biology. We eagerly await your favorable response. Thank you.
sincerely yours,
Sik-Won Choi, Ph.D.

Reviewer 2 Report
Comments and Suggestions for Authors
The authors present a paper detailing the effects of an acorn extract on wound healing. Acorn extracts from this genus have been investigated for their wound healing properties before, however this paper provides a detailed characterisation of a specific species (Quercus glauca) and includes an animal model.
Whilst generally scientifically sound, the authors fail to discuss the limitations of their study; most egregiously they neglect to consider the differences between wound healing in rodents compared to humans. For example, mouse wounds heal by rapid contraction, and not by secondary intention - this is particularly relevant to the discussion of inflammatory modulation and re-epithelialisation.
Additional comments are below:
Line 41: Should read "...human body, and plays..."
Lines 68-69: Please cite the studies at the end of this statement.
Line 124: Ensure "104" is superscripted to read 104
Line 135: Specify the concentrations used, or state "as per section 2.4" - see also for Line 143
Section 2.5: What density were the cells seeded at?
Section 2.6: Please clarify if all samples were treated with H2O2, and justify if/why this treatment was performed after QGASE incubation.
Line 209: Were the calipers only used for the first three days, or at the three time points notated?
Line 213: If these steps are "usual" please cite a paper with these methods.
Line 215: 4mm, or 4 microns?
Lines 219-221: Do you have a rubric that was used to define each score? If so, please provide. Otherwise, please explain how scores were assigned.
Section 2.13: How were p-values adjusted for multiple comparisons?
Line 251: Please cite "previous data"
Figure 3A: Hard to assess figures as quite small. It looks like the area imaged for 1-10μg/mL QGASE has a lower cell density, which would affect MFI calculations.
Line 303: I'm pretty sure the "Figure 1b" here is a typo and should read Figure 4b
Figure 4: Refers to H2O2 "induced" wounds - these wounds don't appear to be caused by H2O2 but are instead a model enhanced by H2O2. Please clarify.
Figure 5C: Please confirm y-axis is "relative fold induction" and not % change.
Authors quantified components of extract by LC-MS and noted some existing literature exploring their antioxidant and wound healing properties. However, I think the authors could better discuss (or speculate on) the potential cellular mechanisms linking the active compounds to the observed outcomes.
Comments on the Quality of English LanguageEnglish language is generally fine with some minor errors.
Author Response
â–ºReviewer 2’s comments:
The authors present a paper detailing the effects of an acorn extract on wound healing. Acorn extracts from this genus have been investigated for their wound healing properties before, however this paper provides a detailed characterisation of a specific species (Quercus glauca) and includes an animal model.
Whilst generally scientifically sound, the authors fail to discuss the limitations of their study; most egregiously they neglect to consider the differences between wound healing in rodents compared to humans. For example, mouse wounds heal by rapid contraction, and not by secondary intention - this is particularly relevant to the discussion of inflammatory modulation and re-epithelialisation.
Additional comments are below:
Line 41: Should read "...human body, and plays..."
- A) Thank you for your suggestion. As recommended, we have revised the sentence.
Lines 68-69: Please cite the studies at the end of this statement.
- A) Thank you for your insightful suggestion. In accordance with your recommendation, we have cited the studies at the end of this sentence, specifically on lines 68-69.
Line 124: Ensure "104" is superscripted to read 104
- A) Thank you for this suggestion. As recommended, we have superscripted to read “104”
Line 135: Specify the concentrations used, or state "as per section 2.4" - see also for Line 143
- A) Thank you for your suggestion. In accordance with your recommendation, we have formatted the text to include a superscript, resulting in the notation
Section 2.5: What density were the cells seeded at?
- A) Thank you for your thorough review. In accordance with your suggestion, we have revised section 2.5 to include a statement regarding cell density.
Section 2.6: Please clarify if all samples were treated with H2O2, and justify if/why this treatment was performed after QGASE incubation.
- A) Thank you for your thorough review. We pretreated QGASE for one hour prior to the H2O2 This pretreatment was performed to enhance the uptake of the compound.
Line 209: Were the calipers only used for the first three days, or at the three time points notated?
- A) We appreciate your suggestion. In accordance with your recommendation, we have revised the sentence to “Every three days, the wound size was measured using a digital caliper”.
Line 213: If these steps are "usual" please cite a paper with these methods.
- A) Thank you for your suggestion. In accordance with your recommendation, we have cited a paper that details the protocols for paraffin block embedding.
Line 215: 4mm, or 4 microns?
- A) We express our gratitude for your thorough review. We have revised the measurement unit to 4 micrometers.
Lines 219-221: Do you have a rubric that was used to define each score? If so, please provide. Otherwise, please explain how scores were assigned.
- A) Thank you for your valuable feedback. We have incorporated the skin wound healing scoring guidelines into the Materials and Methods section, as detailed in Table 2.
Section 2.13: How were p-values adjusted for multiple comparisons?
- A) We calculated p-values using Student's t-test to compare a single group to determine whether the results were statistically significant.
Line 251: Please cite "previous data"
- A) We initially used the term "previous data" to refer to the information presented in However, this phrasing appears to have caused confusion. The descriptions of Figures 1a and 1b have been thoroughly revised.
Figure 3A: Hard to assess figures as quite small. It looks like the area imaged for 1-10μg/mL QGASE has a lower cell density, which would affect MFI calculations.
- A) We appreciate your valuable suggestion. In accordance with your recommendation, we have revised Figure 3A to improve clarity. Furthermore, we conducted measurements of whole cell fluorescence using a fluorescence spectrometer in a 96-well black plate format. As a result, the Mean Fluorescence Intensity (MFI) obtained is consistent with the data presented in Figure 3B.
Line 303: I'm pretty sure the "Figure 1b" here is a typo and should read Figure 4b
- A) We acknowledge that there was an error in the numerical representation, and we appreciate your thorough review. The number has been corrected to Figure 4b.
Figure 4: Refers to H2O2 "induced" wounds - these wounds don't appear to be caused by H2O2 but are instead a model enhanced by H2O2. Please clarify.
- A) Thank you for your thoughtful suggestion. In accordance with your recommendation, we have revised the sentence to “QGASE repaired an H2O2-enhanced wound in HF cells”
Figure 5C: Please confirm y-axis is "relative fold induction" and not % change.
- A) Thank you for your comprehensive review. In accordance with your suggestion, we have modified the unit of the y-axis to reflect relative fold induction.
Authors quantified components of extract by LC-MS and noted some existing literature exploring their antioxidant and wound healing properties. However, I think the authors could better discuss (or speculate on) the potential cellular mechanisms linking the active compounds to the observed outcomes.
- A) Thank you for your valuable suggestion. In alignment with your recommendation, we have elaborated on the health benefits of polyphenolic compounds to underscore the significance of our observed results in the Discussion section.
We would like to express our sincere gratitude to you and the reviewers for your valuable feedback. We hope that the revised version of our manuscript is now deemed suitable for publication in Biology. We eagerly await your favorable response. Thank you.
sincerely yours,
Sik-Won Choi, Ph.D.
